

# Evaluation of Greenland near surface air temperature datasets

J. E. Jack Reeves Eyre[1] and Xubin Zeng[1]

[1]Department of Hydrology and Atmospheric Sciences, University of Arizona, Tucson, 85721, USA

*Correspondence to*: J. E. Jack Reeves Eyre (jeyre@email.arizona.edu)

**Abstract.** Near-surface air temperature (SAT) over Greenland has important effects on mass balance of the ice sheet, but it is unclear which SAT datasets are reliable in the region. Here extensive in-situ SAT measurements (~ 1400 station-years) are used to assess monthly mean SAT from seven global reanalysis datasets, four gridded SAT analyses, one satellite retrieval and two dynamically downscaled reanalyses. Strengths and weaknesses of these products are identified, and their biases are found to vary by season and glaciological regime. MERRA2 reanalysis overall performs best with mean absolute error less

than 2 $^{\circ}$C in all months. Ice sheet-average annual mean SAT from different datasets are highly correlated in recent decades, but their 1901–2000 trends differ even in sign. Compared with the MERRA2 climatology combined with gridded SAT analysis anomalies, thirty-one earth system model historical runs from the CMIP5 archive reach ~5 $^{\circ}$C for the 1901–2000 average bias and have opposite trends for a number of sub-periods.

## 1 Introduction

Near-surface air temperature (SAT) over the Greenland ice sheet (GrIS) is important both for its place in wider climate change and for its effects on mass balance of the ice sheet. Due to its remoteness and extreme climate however, continuous widespread climate monitoring over the GrIS has been carried out for only about the last two decades, and even then with rather sparse coverage in some geographic areas and glaciological regimes. Studies of past climate and surface mass balance (SMB) of the GrIS have used a variety of techniques to achieve complete spatial coverage of SAT, including statistical

interpolation, atmospheric reanalysis, dynamic downscaling through regional climate modeling, and satellite remote sensing. Projections of future change in Greenland climate and ice sheet evolution have used global earth system models, either directly (e.g., Ridley et al., 2005; Vizcaíno et al., 2013) or through dynamical downscaling (e.g., Fettweis et al., 2013; Rae et al., 2012). Many such studies have involved some form of assessment using weather station data (e.g., Box, 2013; Noël et al., 2015; Rae et al., 2012) and inter-comparison of several SAT data sources (e.g., Box, 2013). Here we build on such work

to assess and compare a greater number of widely available products, using a more comprehensive set of in situ observations than has customarily been used in previous work. In doing so we hope to guide future dataset and model development over this region and address a number of outstanding questions.

Our main focus here is on global datasets – reanalyses, gridded SAT analyses and earth system models from the CMIP5 archive – though two regional datasets are also included. Regional climate models (RCMs) have been used widely to





downscale reanalysis (e.g., Box, 2013; Box et al., 2009; Burgess et al., 2010; Ettema et al., 2010a; Noël et al., 2015) and global climate model output (e.g, Fettweis et al., 2013b; Rae et al., 2012). While Noël et al. (2016) demonstrated the benefit of high (< 10 km) resolution for SMB, the benefit for SAT is less clear: Lucas-Picher et al. (2012) showed that SAT bias is not reduced by changing the grid size from 0.25° to 0.05°. By comparing results from a range of resolutions, including RCMs

at relatively high resolutions, we aim to investigate the value added by dynamic downscaling.

      Inter-comparison of SMB components has been carried out among different RCMs and between RCMs and global reanalyses (Cullather et al., 2016; Rae et al., 2012; Vernon et al., 2013). The results from these studies point to a wide inter-model spread, which the authors related to differences in model parameterizations (e.g., snow and ice physics), model ice mask and forcing at the domain lateral boundaries. One goal of this work is to investigate how closely RCM forcing affects

SAT representation, by comparing differently forced runs of the same RCM, and comparing these runs with results taken directly from the forcing dataset.

      Satellite remote sensing data has been key in spatially complete reconstruction of GrIS SAT, whether through direct use (e.g., Hall et al., 2013) or through assimilation into reanalyses. One consequence of this, though, is that only a small proportion of studies extend GrIS SAT back before the satellite era. SMB studies that incorporate centennial scale SAT

reconstructions include: Hanna et al. (2011), who combined Twentieth Century Reanalysis (Compo et al., 2011) and ERA–40 reanalysis (Uppala et al., 2005); and Box (2013) who adjusted regional climate model output using in situ observations to reconstruct SAT from 1840–2010. The SAT reconstruction was compared to that of Hanna et al. (2011) and found to be cooler over most of the common period, but especially so before about 1930. By looking at multiple datasets that include the first half of the 20[th] century (and earlier), we hope to shed light on the climate of the GrIS in this very poorly observed

period. In particular, such datasets allow comparison with previous assessments of Greenland SAT climate based on (mainly coastal) station data (e.g., Box, 2002; Chylek et al., 2006; Hanna et al., 2012; Mernild et al., 2014). Long, spatially complete time series also offer the best means of assessing CMIP5 models, without differences introduced by incomplete spatial coverage and short period (~ 30 year) trends and decadal variability.

      This paper is structured as follows: in Sect. 2 data sources are described and examples of their past use given;

results are broken down into Sect. 3.1, dataset assessment using in situ observations, Sect. 3.2, comparison of long-term SAT changes among datasets and Sect. 3.3, further discussion; conclusions are presented in Sect. 4.

## 2 Data

### 2.1 Weather station observations

      To assess the different SAT products, we use SAT observations, made at manned and automatic weather stations

(AWSs) from several sources, totalling 17000 station-months or 1400 station-years. These are briefly described here, and further details are shown in Fig. 1. Coastal station records of monthly mean temperature for 11 stations (stretching as far back as 1784) are compiled by the Danish Meteorological Institute (DMI; Cappelen, 2014). Thanks to their long records,



SAT from these stations has been studied extensively: Box (2002) found a pattern of warming from ~1900 to ~1940, cooling from ~1940 to ~1990, and warming from ~1990 onwards. In addition, inter-annual variability was found to be closely related to the North Atlantic Oscillation (NAO). Hanna et al. (2012) found similar patterns of warming and cooling using updated SAT data from DMI stations, and concluded that recent temperatures were in excess of SAT from the early 20[th] century

warm period.

In contrast to coastal regions, no long term (e.g., 30 years or more) climate monitoring has occurred on the GrIS. Monthly mean temperatures from mid-20[th] century expeditions and field camps, concentrated in the 1930s and 1950s, are taken from the appendix of Ohmura (1987). Since the mid-1990s, the number of SAT observations from the ice sheet has greatly increased. We use records from AWSs operated as part of the Greenland Climate Network (GC–Net), predominantly

in the accumulation region of the ice sheet (Steffen and Box, 2001), from the K–transect in western Greenland (operated by the Institute for Marine and Atmospheric Research at the University of Utrecht; van de Wal et al., 2005) and from AWSs mostly in the ablation region operated by the Geological Survey of Denmark and Greenland (GEUS) under the PROMICE and GAP programs (Van As et al., 2011). Locations and types of all stations are shown in Fig. 1 and further details are available in the Supplemental Material (Table S1).

The providers of several of these observational datasets employ quality control tests and/or quality inspection as part of their routine data management. In addition, we remove unrealistic values where our inspection of time series reveals them (e.g., with spikes and step changes). Where data were provided as hourly values, we calculate daily averages (the mean of hourly values) for all days with 20 or more hourly values and monthly averages (the mean of daily values) for all months with 24 or more daily values.

**2.2 Gridded SAT products**

Most of the datasets assessed here fall into two categories: global reanalysis and interpolated global SAT analyses. The spatial and temporal resolution and length of record (Table 1) vary greatly across these products. It should be noted that even though reanalyses are constrained by (in some cases) remote sensing and some local observations to represent observed synoptic–planetary scale weather, the lack of assimilated SAT observations over Greenland means that the SAT data

assessed here are largely the result of modelled atmospheric and surface processes.

Several of the latest generation of global reanalyses are used in this study (Table 1). Most of these are heavily reliant on radio-sonde and satellite data, and thus cover only the period when these are available (1979 onwards; 1958 in one case). In addition, we analyze the Twentieth Century reanalysis (20CR; Compo et al., 2011) and ERA–20C (Poli et al., 2016), which do not assimilate satellite or radio-sonde data, but instead use a subset of observation types that are available over the 20[th]

century (and earlier) and therefore cover much longer periods. GrIS SAT from reanalyses has been used in SMB modeling: Hanna et al. (2005) used ERA–40, while Hanna et al. (2011) combined ERA–40 with 20CR. However, SAT data from a number of other reanalyses remain untested for such applications.





Reanalysis represents a combination of observations and model. In contrast, several research groups have created gridded SAT datasets based almost entirely on statistical analyses of weather station SAT (we refer to these as *gridded SAT analyses*). Such datasets have not been widely used over Greenland (though see, e.g., Fettweis et al., 2008), and their long time series and temporal homogeneity is a potential strength. For example, some reanalyses are known to suffer from

spurious trends as observing networks and processing systems change (e.g., Screen and Simmonds, 2010): comparison between reanalyses and gridded SAT analyses, particularly in the early 20$^{th}$ century, can highlight such problems with reanalyses. Some gridded SAT analyses, due to their analysis methods and requirements for data completeness, have large data gaps over Greenland, e.g., HadCRUT4 (Morice et al., 2012) and NOAAGlobalTemp (Smith et al., 2008; Vose et al., 2012). However, we here use four such datasets that have complete (or very nearly so) coverage over Greenland (Table 1).

Note that GISTEMP (Hansen et al., 2010) is provided as anomalies only (relative to 1951–80 climatology). As the ice sheet weather stations have typically not been operational long enough to calculate a stable climatology, we do not assess GISTEMP using in situ observations; however, we do combine GISTEMP anomalies with MERRA2 climatology to enable comparison of long-term variability against other datasets.

Recognizing that reanalysis SAT over Greenland is dominated by the model formulation and has relatively coarse

horizontal resolutions, a number of researchers have sought to improve results over the GrIS by using reanalysis to force higher resolution regional climate models (RCMs) coupled to comparatively sophisticated snow–ice models. Such models are typically run with grid spacing of around 10–20 km. This high resolution (compared to global climate models and most reanalyses) is thought to better resolve the large climate gradients that occur around the margins of the ice sheet. Here we include output from version 3.5 of the Modèle Atmosphérique Régional (MAR; Fettweis et al., 2013b) run with 20 km grid

spacing, then interpolated to the 5 km polar stereographic grid of Bamber et al. (2001). Two different runs of MAR are used here: one forced by ERA–40 (1958–1978) and ERA–Interim (1979–2015) reanalyses; the other forced by 20CR reanalysis. ERA–40 and ERA–Interim reanalyses have been widely used as forcing data (Box et al., 2009; Ettema et al., 2010a, 2010b; Fettweis et al., 2013); 20CR has not. It should be noted that the field we use from this model is nominally the 3-meter air temperature, whereas most reanalyses output 2-meter air temperature (when specified), and the measurement height at

weather stations varies as the snow/ice surface changes.

Satellite remote sensing data, in addition to being assimilated by reanalyses, have been used directly to study the GrIS. Several studies have focused on the relationship between SAT and ice sheet surface temperature (IST), and have used data from both microwave (e.g., Shuman et al., 1995, 2001) and infrared sensors (e.g., Comiso et al., 2003; Hall et al., 2008, 2013; Koenig and Hall, 2010). Sounding instruments offer a method to retrieve air temperature more directly, but have

received little attention over GrIS. Here we assess SAT from the Atmospheric Infrared Sounder (AIRS; Chahine et al., 2006) on board NASA's AQUA satellite platform. AIRS has been operational since September 2002, providing temperature and humidity retrievals at many vertical levels through the atmosphere. We use the level 3 monthly near surface air temperature from ascending and descending overpasses, taking a weighted average to give a single monthly value at each grid point (further details are given in Table 1). This product is a clear-sky only retrieval: a key part of assessing this product is to





understand what effect this has through, for example, seasonally varying cloud amounts and increased wind-driven mixing during winter storms, as discussed in Koenig and Hall (2010).

Earth System Models (ESMs) from the CMIP5 multi-model ensemble archive (Taylor et al., 2011) are not assessed using in situ observations but are included in comparisons of long term areal average SAT. Comparison against in situ
observations is not performed because the ESMs are free-running coupled (atmosphere–ocean–land–ice) models, so we do not expect them to have the correct phasing of synoptic weather or inter-annual or even decadal climate. Apparent biases at station locations would therefore combine bias in the long term average and differences in variability over the relatively short station records. The ice sheet areal averages, compared to the longer reanalyses and gridded SAT analyses, should adequately reveal the first order biases in the ESMs' long term average SAT and its trends. Thirty-one different model
configurations, from 11 modelling centers are used. We use the first ensemble member (r1i1p1) only of historical runs. Further details of individual models are given in Table S2. In contrast to other datasets above, CMIP5 ESM SAT data are used on their model native grids, rather than interpolated to a common grid (to be discussed below).

## 3 Results

Our analysis is based on the monthly mean near-surface air temperature. Except for CMIP5 ESMs and the MAR RCM
variants, datasets were spatially interpolated from their native grid to a 5 km equal area grid (the EASE grid of the National Snow and Ice Data Center [NSIDC]) using bilinear interpolation. This resolution is used to attempt to resolve the large SAT gradients that occur over the steep topography at the margin of the ice sheet. Interpolating like this presents some potential problems due to model topography: The surface elevation fields used in many of the datasets here are smoother than the actual topography of Greenland, and this leads to elevation biases as seen in Fig. 2. The relatively low resolution 20CR (Fig.
2b) has mostly positive elevation bias around the edge of the ice sheet, and negative bias in the interior. The higher resolution MAR (Fig. 2c) does not have the same magnitude of biases in the interior, but still misses much of the small scale detail, as seen by the speckled pattern of biases of alternating sign. All datasets have a negative mean elevation bias on the ice sheet (Table 2), with MAR the smallest and 20CR the largest. Note that elevation errors are not a monotonic function of resolution: despite a smaller grid spacing than MERRA2 and ERA–Interim, CFSR still has a larger bias and mean absolute
error.

The elevation biases cause the SAT fields to be smoother than in reality, and interpolation of the smooth SAT fields is unlikely to accurately reflect the true SAT gradients, which are strongly influenced by elevation. To account for this, a correction is applied to the reanalysis and AIRS datasets after interpolation to the EASE grid: for each product, the elevation field is also bilinearly interpolated to the EASE grid, and then compared to the digital elevation model (DEM) of Bamber et
al. (2013; provided at 1 km grid spacing, and here bilinearly interpolated to the EASE grid). The elevation bias (product minus DEM) is multiplied by the relevant month's lapse rate from Fausto et al. (2009) and their product added to the





interpolated SAT field. The importance of this step can be seen by comparing the results below with comparable figures for un-corrected datasets (Figs. S1 and S2).

**3.1 Monthly mean SAT biases**

Comparisons between gridded datasets and in situ observations are made by choosing the nearest EASE grid point. Note that
an alternative, using bilinear interpolation directly from the native grids to the station locations, gives very similar results. The primary statistics used in the assessment of datasets are mean bias and mean absolute error (MAE). When aggregating results over multiple stations, the average of station-months is taken, rather than averaging over time then over stations. Stations are grouped into coastal (DMI), ice sheet below 1500 m and ice sheet above 1500 m. The elevation of 1500 m is chosen to approximately represent the equilibrium line altitude, as found for the K–transect by van de Wal et al. (2005). The
pattern of biases seen below is largely the same for different separation elevations between 1000 m and 2000 m. Note that, when taking the spatial average across the ice sheet, the area above 1500 m dominates: using the DEM and mask of Bamber et al. (2013), Greenland has a total area of 2.16 million km$^2$, which is 16.5 % ice-free land, 18.6 % ice sheet below 1500 m, and 64.9 % ice sheet above 1500 m.

Before looking at results aggregated across many stations, we briefly present time series from a few individual
stations. Figure 3 shows the marked SAT differences from two datasets in comparison to in situ observations. MERRA2 closely matches the observed seasonal cycle at all stations, while CRU has biases that vary with season and in some months reaches ~10 $^o$C. Close inspection reveals differences in CRU biases between Summit (large biases in winter, small in summer) and the lower elevation stations (large biases in both winter and summer). The patterns of bias at the three lower elevation sites are similar – independent of observing network and therefore instrument type. This suggests that the
difference with Summit represents genuine spatial variation of bias, rather than instrument effects.

To generalize these results to a larger number of stations and longer time period, Figs. 4 and 5 show the seasonal cycle of bias and MAE averaged over all station months from 1979 onwards. Many datasets, though not all, show similar seasonal cycles: above 1500 m and at coastal stations, more positive biases in winter and more negative in summer; at ice sheet stations below 1500 m, the opposite cycle. Despite qualitative similarities, a clear picture of dataset performance
emerges. On the ice sheet, MERRA2, MAR (both versions) and 20CR are best. The AIRS satellite product is also very good, except in winter months at stations below 1500 m. At coastal stations, ERA–Interim performs best, and JRA–55 and MERRA2 are nearly as good; MAR (both versions) performs well in summer, but poorly in winter. Note that without elevation corrections, MAR coastal station errors are larger in summer but smaller in winter (Figs. S1 and S2). CRU and Berkeley Earth results are comparable to the best reanalyses at coastal stations, likely because it is SAT observations from
the coastal stations that form the majority of the input data for these datasets. Based on a (rather subjective) assessment of the 12-month average bias (absolute value, to avoid cancellation between months) and MAE, the most consistent good performer is MERRA2: the 12-month average biases (absolute values) are approximately equal or less than 1.0 $^o$C, and 12-





month average MAE are less than 1.5 °C, in all regions. MAR–ERA and MAR–20CR have comparable performance across the ice sheet, but overall are marred by large winter time biases at coastal stations.

The analysis above aggregates all station months from 1979 onwards. To investigate time variations in biases, Fig. 6 compares mean bias before and after 1979 for those datasets which begin before 1979. Note that the datasets beginning in
1979 show only small changes in bias by decade (not shown). GISTEMP is included here with the MERRA2 elevation-corrected climatology: the absolute values of the biases are highly dependent on the climatology, but here can be ignored as we are interested in the *changes* in bias.

Clear differences are apparent for some seasons and datasets. Statistical significance of these differences (using Student's *t*-test for a difference in means with unequal variances, and defining significance at the 1 % level) suggest that a
number of the datasets have time-varying biases. This is most apparent for the coastal DMI stations, where larger sample sizes give the statistical test greater power. 20CR has negative changes high on the ice sheet and at coastal stations but positive changes in the ablation region. NANSENSAT shows negative changes over time in all regions in both winter and summer. Other than NANSENSAT, the gridded SAT analyses do not seem more prone to time-varying bias than reanalyses.

**3.2 Time series**

Areal average (weighted by glacial ice fraction) annual mean temperatures for all datasets show close correlation in recent decades: considering only the period 1979 onwards, the correlation (*r*) values are in the range 0.71 to 0.99. In earlier periods, correlations are generally smaller: for the period 1900–1940, we have *r* = 0.34 to 0.96, and for 1940–1980, *r* = 0.28 to 0.95. However, CRU, Berkeley Earth and GISTEMP have pairwise correlation coefficients of 0.90 or greater for all these periods
since they are based on a similar set of surface stations. 20CR is highly correlated (*r* > 0.90) with MAR–20CR in all these periods as the latter is driven by the former.

Among the datasets covering the entire 20[th] century, most have similar inter-decadal variations, with a general pattern of early 20[th] century warming, up to 1930, followed by cooling to around 1990, then strong warming in recent years (Fig. 7). Nonetheless, differences do exist (Table 3). For instance, NANSENSAT shows relatively large early 20[th] century
jumps thought to be caused by changing data sources over this period, indicating this dataset is not suitable for long-term monitoring over Greenland. In 20CR, the ~1930 peak is warmer than the most recent years, in contrast to other datasets. Related to the last point, the amount of cooling from 1930 to 1990 varies between datasets, with 20CR showing strongest cooling, and GISTEMP showing least. Anomalies (relative to the 1981–2010 mean; Fig. 7b) reveal some more subtle differences. For example, MAR–ERA and CRU both show less positive anomalies than other datasets since about 2005.
Variability in 20CR matches other datasets closely from 1980 onwards, but before this differs significantly (except in the 1940s). MAR–20CR shows agreement for more of the period, but seems to inherit poor representation of variability before about 1920 and from 1950 to 1980.





Of the datasets that extend back before 1900, Berkeley Earth and GISTEMP agree quite closely but show notable differences with 20CR. Berkeley Earth and GISTEMP cannot be considered independent data sources (as they use similar input data for this period), and so their consensus is not especially meaningful. However, the fact that their biases are more constant in time (Fig. 6) than those of 20CR suggest that they are more reliable for this period. In common with disparities
mentioned above for the first half of the 20[th] century, users of these SAT datasets should be aware that significant uncertainties exist before 1900, with notable differences in trends and variability (both inter-annual and inter-decadal). We recommend the use of gridded SAT analyses alongside reanalyses and downscaled reanalyses, to assess sensitivity to these differences.

The range of SAT among CMIP5 ESMs is wider than that among the other datasets (Fig. 7), but much of this range
comes from a group of four relatively warm models and two relatively cold models: eliminating these gives a range comparable to the gridded analysis and reanalysis datasets. This highlights the fact that choice of verification dataset can have a significant effect on assessments of ESM mean climate. Based on results above, we use GISTEMP with MERRA2 climatology to assess the long-term mean temperatures of the CMIP5 ESMs. Using the 1901–2000 mean of ice sheet annual average temperatures, 10 ESMs lie within 1 °C (namely GFDL's CM3 and ESM2G; GISS's E2–H–CC, E2–R and E2–R–
CC; IPSL's CM5A–MR, CM5A–LR_historical and CM5A–LR_esmHistorical; CESM1–CAM5; CMCC–CMS; see Table S1 for further details).

The median of the CMIP5 ESM trends (Table 3) is positive for all periods considered – in marked contrast to the other datasets. However, further investigation shows the picture is not so clear: The number of individual ESMs that have positive trends in each period suggest that, with the possible exception of 1990–2005, the models do not give a clear
consensus on signs of trends: this may be because inter-decadal climate variability dominates, and the phasing of this variability differs between models. For the 1990–2005 period, 27 out of 31 ESMs have a positive trend and the median is an order of magnitude larger than for the earlier periods (although still smaller than the 1990–2005 trends from the other datasets). Thus the ESMs seem to agree on accelerated warming since 1990. Significance of the trends is tested using the method described in Santer et al. (2000), which is based on a two-tailed Student's *t*-test modified to account for
autocorrelation in the time series. Few of the trends are significant, in any of the periods considered. The ensemble mean has a long term average slightly higher than that from MERRA2+GISTEMP, and trends broadly similar to the median of the individual trends (i.e., with accelerated warming since about 1970); however, it does not feature decadal variability that individual CMIP5 ESMs, reanalyses and gridded SAT analyses show, and thus has limitations in representing historical GrIS SAT.

**3.3 Further discussion**

The majority of in situ SAT observations from the ice sheet have been made since 1995. We have used the relatively small number of observations from the mid-20[th] century to assess the stationarity of biases, and find that several datasets show significant temporal variations in their bias. At ice sheet stations above 1500m (the region which dominates in areal





averages), 20CR shows large (and significant) changes – becoming more negative with time. 20CR biases also become more negative with time at coastal stations (from which there are many more observations), casting further doubt on the suitability of this dataset for long-term trend analysis. ERA–20C has more stable biases in the accumulation region and at coastal stations (though not in the ablation region), as do several of the gridded SAT analyses, suggesting that SAT reconstruction

based on anomalies is valid over monthly to centennial time scales. This is not a trivial result, as it is not obvious a priori that conditions driving anomalies at coastal stations will result in a similar, smoothly varying response over different surface types.

Trends among the datasets assessed here (excluding CMIP5 ESMs) generally agree with patterns found in previous studies (e.g., Box, 2002). In addition, interannual variability since 1979 matches closely between datasets. However,

differences between longer term trends, along with temporal changes in bias (discussed above), suggest that some datasets have limitations in their representation of early to mid-20th century GrIS SAT. In particular, 20CR shows stronger cooling between 1930 and 1990 than most other datasets, and has a 1930s warm period warmer than the 21st century warm period. Such discrepancies between 20CR and anomaly based SAT datasets have been noted at the global scale by Compo et al. (2013), although the differences here are much greater than those for global SAT. Similarity of anomalies among gridded

SAT analyses and ERA–20C, along with the greater temporal constancy of their biases, leads us to put greater faith in their representation of long term trends and inter-decadal variability.

One of our central questions in this study is whether global SAT datasets are as good as RCM-downscaled datasets, which are, at least for SMB modeling, the current state of the art. For MAR–ERA, results are generally better than for SAT taken directly from the forcing dataset (even with elevation corrections applied). However, at coastal stations, MAR–ERA

performs worse than ERA–Interim. For MAR–20CR, the difference is minimal at ice sheet stations and downscaling is detrimental at coastal stations in winter (though without elevation corrections, MAR–20CR has smaller biases and MAE than 20CR; see Fig. S2). Comparing against all global datasets, we find MERRA2 has biases and MAE comparable to or less than MAR (both forcings) in all seasons and regions. This is likely due to the comprehensive (relative to other reanalyses) snow/ice model in MERRA2 (Cullather et al., 2014) and reinforces the importance of atmosphere–ice sheet coupling in

modeling SAT. Thus the benefits for SAT of RCM downscaling are not clear.

Another question related to the RCM downscaling is: how closely does the forcing dataset constrain climate variability in the downscaled? Correlations (of ice sheet annual mean SAT) before 1979 suggest that the constraint is close: MAR–20CR has correlation coefficients with 20CR greater than 0.9 for both 1900–1940 and 1940–1980, while its correlation with other datasets is lower (0.54–0.62 for 1900–1940; 0.29–0.81 for 1940–1980). Consideration of anomalies

(Fig. 7b) suggests that the downscaling improves representation of climate variability by bringing MAR–20CR more into line with other datasets. Nonetheless, differences remain, particularly before 1920 and between 1950 and 1980, and we consider that MAR–20CR still suffers from some shortcomings in 20CR's representation of variability before 1980.

Although the comparison is for a shorter period than for other datasets, we have found that AIRS gives very good results over the ice sheet in summer – with smaller biases and MAE values than any other dataset in the ablation region for





June, July and August. However, its performance is poor in winter over the ablation region and in summer at coastal stations. The wintertime biases in the accumulation region do not agree (although those in the ablation region do) with the findings of Koenig and Hall (2010) at Summit, that satellite-derived clear-sky only temperatures were lower than all-cloud in situ measurements. They attributed this finding to the fact that clear-sky only retrievals miss winter storms – during which strong

winds mix warm air from above an inversion down to the surface. The summertime results suggest AIRS may be a useful dataset for studies of recent SMB, but further investigation is needed into the consequences of clear-sky retrievals, particularly the wintertime discrepancy with previous work and the possibility of compensating errors in summertime.

Note that there is a discrepancy between various products in calculating monthly mean SAT. As discussed in Wang and Zeng (2013) the daily mean calculated using 24 hourly values per day is different from that calculated using just

maximum and minimum SAT. Comparisons for AWSs on the GrIS suggest the difference for monthly mean temperatures is ~ 0.2 $^{o}$C, but can exceed 0.5 $^{o}$C in some individual months. Other averaging methods (e.g., mean of 3-hourly values; weighted mean of 0800, 1400 and 2100 local time (Box, 2002)) are unlikely to introduce larger errors than the maximum plus minimum method. Overall these relatively small uncertainties are unlikely to affect our conclusions.

Our evaluation of 5 km grid box values using point measurements may also be affected by the sampling errors due

to the SAT variation within a grid box (e.g., in grid boxes containing a large range of elevations and different surface types). Quantifying such an error could in principle be done using several stations within the same grid box; we do not have any 5 km grid boxes containing more than one station, however. Instead we look to the variation of elevation, assuming that this is the dominant source of SAT variation at small spatial scales and implicitly neglecting effects of varying surface type and other factors. Elevation variation at any particular location is quantified by taking the standard deviation of elevation values

at the nearest and 24 surrounding grid boxes from the 1 km version of the Bamber et al. (2013) DEM. This is then multiplied by a (slightly conservative) lapse rate of 9.0 $^{o}$C km$^{-1}$, to give a likely range of SAT variation over this elevation range. This formulation gives smaller sampling error over relatively flat terrain: ~ 0.1 $^{o}$C above 1500 m on the ice sheet. In more variable terrain, around the margins of the ice sheet and in coastal land regions, sampling errors are larger – usually in the range 0.3 to 1.0 $^{o}$C. Overall these uncertainties are relatively small in magnitude compared to the large biases and MAEs between

various datasets and in situ observations, and hence our conclusions are largely unaffected.

In our assessment of biases and their changes through time we have assumed that all observations are un-biased. Observation biases are likely to exist (e.g., the positive bias of un-aspirated thermometer shields in low wind, high solar radiation conditions; (Genthon et al., 2011) and are likely to vary in space and time due to differences in station siting, instrumentation and observing practices (e.g., number per day and timing of manual thermometer readings). By breaking

down the bias assessment into two altitude bands, our analysis aims to reduce the impact of station siting changes (e.g., a large increase in the proportion of ablation zone observations as the PROMICE network has been set up). Our analysis also, to some extent, isolates different instrument types, as the PROMICE network and K–transect stations are mostly below 1500 m, while GC–Net stations are mostly above 1500 m. Side-by-side comparisons of different instrument types, across different climatological regimes on the ice sheet, is needed for a future study to better understand the spatial and temporal patterns of





bias shown here. This could include the replication of historic observing practices and instruments, to better understand, and make the most of, the limited number of mid-20[th] century ice sheet SAT observations.

## 4 Conclusions

We have assessed a number of global SAT datasets using in situ observations over Greenland, and found large differences in
their performance. Reanalyses generally perform better than gridded SAT analyses – particularly at high elevations on the ice sheet. Simple elevation-based corrections applied to reanalyses lead, in most cases, to improved performance. Considering all regions and seasons, the smallest biases are seen in (elevation-corrected) MERRA2 reanalysis. Biases vary by season and by region of the ice sheet: in the ablation region (demarcated here by the 1500 m elevation contour) during summer, most reanalyses have a ~1 $^{o}$C positive bias (though 20CR and ERA–20C have negative biases) while CRU and
Berkeley Earth gridded SAT analyses have larger positive biases. These biases have implications for SMB reconstruction, as this region and season contribute a large proportion of meltwater creation.

Among global datasets that cover the entire 20[th] century, 20CR generally has the smallest biases and MAEs when comparing against observations made since 1979. However, combining GISTEMP anomalies with the MERRA2 climatology gives slightly better results and, given concerns about spurious long term trends in 20CR (in particular, a warm
bias before 1950), we recommend this type of approach (i.e., combining GISTEMP with MERRA2) to represent monthly SAT over the early and mid-20[th] century. Similarity of anomalies between gridded SAT analyses (except NANSENSAT) suggests that observed biases result from their climatology fields, but their anomalies are suitable alternatives to GISTEMP.

Alongside multi-decadal global SAT datasets, we have analyzed SAT from recent (2002 to present) AIRS satellite retrievals and from RCM-downscaled reanalysis. AIRS has among the smallest biases and MAE in summer months over the
ice sheet, but larger errors in winter and when comparing to coastal stations. RCMs are found to reduce biases in comparison to their respective forcing datasets and provide among the best representations of SAT on the ice sheet. However, MERRA2 reanalysis performs comparably on the ice sheet, and better in comparison to coastal stations. The long-term variability of RCM SAT is found to be strongly constrained by the forcing dataset; the shortcomings that we highlight for 20CR thus also persist, to some degree, in the version of MAR forced by 20CR.

We have assessed CMIP5 ESMs by comparing their ice sheet average SAT with that from other datasets. A key finding is that such an assessment depends crucially on the choice of verification dataset. Using GISTEMP combined with MERRA2 climatology (due to its overall good performance in comparison with in situ observations), we find that a large number of the CMIP5 ESMs have similar ice sheet long-term annual average SATs (10 within 1 $^{o}$C, 19 within 2 $^{o}$C). The 1901–2000 trends from most individual models and the ensemble mean are positive. For a number of sub-periods examined,
some individual ESMs have negative trends, though the ensemble mean does not, highlighting the fact that the ensemble mean does not exhibit realistic decadal variability. The 1990–2005 trends are positive and larger than for earlier periods



(though mostly not statistically significant) for the majority of CMIP5 ESMs analyzed here, suggesting that forced changes dominate over internal variability in this period.

Our analysis highlights several avenues for future work. Comparison of different instrument types and measurement practices would allow a quantitative assessment of the effects of instrument bias on the results shown here. Such work is also

crucial to investigations of GrIS diurnal temperature variation, for example in model assessment and SMB studies using positive degree day methods (Fausto et al., 2011; Rogozhina and Rau, 2014). Results for AIRS retrievals suggest it may provide useful SAT information over the GrIS in summer, but further work is needed on the effects of only sampling clear-sky SAT. Investigation is required to establish the cause of disparities in trends and variability between 20CR and ERA-20C – which are ostensibly formulated in similar ways. Possible causes include different representation of atmospheric

circulation and different sea ice and sea surface temperature datasets. While RCM downscaling is currently an important tool in assessing past and future GrIS mass balance changes, our results provide new evidence that results from RCMs are highly dependent on the forcing. The greatest SAT differences between the two versions of MAR used here occur before 1980, but there are differences since 2000 too. Finally, we suggest that more thorough investigation of RCM forcing effects, including spatial distribution of differences and other variables, may be illuminating in reconstructing past GrIS climate and SMB.

**5 Code and data availability**

Most of the data used in this work are freely and publicly available. Full dataset references are given in the Supplemental Material. Derived data fields (e.g., elevation-corrected SAT) and code used to analyze data and plot figures are available from the corresponding author on request.

**Acknowledgements**

This research was supported by NASA (NNX14AM02G), DOE (DE-SC0016533), and the Agnese Nelms Haury Program in Environment and Social Justice. We also thank the various groups and centers for making their datasets and model results available. We thank C.J.P.P Smeets and the Institute for Marine and Atmospheric Research at the Utrecht University for providing the K-transect data. DMI AWS data were downloaded from http://www.dmi.dk/laer-om/generelt/dmi-

publikationer/2013/, GC-Net AWS data from http://cires1.colorado.edu/steffen/gcnet/, and PROMICE data from http://www.promice.dk/. MERRA, MERRA2 and AIRS data were downloaded from the NASA Goddard Earth Sciences Data and Information Services Center (GES DISC). ERA–Interim and ERA–20C were downloaded from the EMCWF website (http://apps.ecmwf.int/datasets/). JRA-55, CFSR and CFSv2 data were obtained from the Research Data Archive at the National Center for Atmospheric Research (NCAR) Computational and Information Systems Laboratory. 20CR data

were downloaded from http://www.esrl.noaa.gov/psd/. CRU data were downloaded from the British Atmospheric Data




Centre (BADC), Berkeley Earth from http://berkeleyearth.org/data/, GISTEMP from http://data.giss.nasa.gov/gistemp/, and NANSEN SAT from http://www.niersc.spb.ru. MAR data were downloaded from ftp://ftp.climato.be/fettweis/MARv3.5.2/Greenland/. CMIP5 data were obtained from the U.S. Department of Energy's Program for Climate Model Diagnosis and Intercomparison (http://cmip-pcmdi.llnl.gov/cmip5/data_portal.html).

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



**Table 1: Temperature products assessed in this work. Latitude longitude spacing refers to the grids downloaded for this work (not necessarily the native model grid). Maximum output frequency refers to the maximum available – monthly averages are used in the analysis.**

| Type | Dataset | Center | Latitude longitude spacing [a] | Maximum output frequency | Period | Reference |
|---|---|---|---|---|---|---|
| *Reanalysis* | MERRA | NASA/ GMAO | 0.5º x 0.667º | Hourly | 1979–2015 | Rienecker et al., 2011 |
| | MERRA2 | NASA/ GMAO | 0.5º x 0.625º | Hourly | 1980–2015 | Molod et al., 2015 |
| | CFSR and CFSv2 [b] | NCEP | 0.5º x 0.5º | Hourly | 1979–2015 | Saha et al., 2010; Saha et al., 2014 |
| | 20[th] Century Reanalysis V2c | NOAA/ CIRES | ~1.9º x 1.875º | 3–hourly | 1851–2014 | Compo et al., 2011 |
| | ERA–Interim | ECMWF | 0.75º x 0.75º | 3–hourly | 1979–2015 | Dee et al., 2011 |
| | ERA–20C | ECMWF | 1º x 1º | 3–hourly | 1900–2010 | Poli et al., 2016 |
| | JRA–55 | JMA | ~0.56º x ~0.56º | 3–hourly | 1958–2014 | Kobayashi et al., 2015 |
| *Gridded temperature analysis* | GISTEMP | NASA/ GISS | 2º x 2º | Monthly | 1880–2015 | Hansen et al., 2010 |
| | CRU TS 3.23 | CRU | 0.5º x 0.5º | Monthly | 1901–2014 | Harris et al., 2014 |
| | Berkeley Earth Surface temperature | Berkeley Earth | 1º x 1º | Monthly | 1750–2016 | Rohde et al., 2013 |
| | NANSENSAT | Nansen Centers | 2.5º x 2.5º | Monthly | 1900–2008 | Kuzmina et al., 2008 |
| *Satellite* | AIRS | NASA | 1º x 1º | Monthly | 2002–2015 | Chahine et al., 2006 |
| *Regional down-scaling* | MAR–ERA | University of Liège | 5 km x 5 km [c] | Monthly | 1958–2015 | Fettweis et al., 2013 |
| | MAR–20CR | | | | 1900–2010 | |

[a] as downloaded for this study; [b] CFSR, covering 1979–2010, and CFSv2, covering 2011–2015, are appended and referred to together as CFSR in the text; [c] Both MAR variants are on the polar stereographic grid of Bamber et al. (2001).



**Table 2:** Error statistics of model elevation fields (interpolated to EASE grid, except for MAR) relative to the digital elevation model (DEM) of Bamber et al. (2013). Bias and deciles are calculated as (model minus DEM). Averages are taken over all ice sheet grid points, classified using the mask of Bamber et al. (2013).

| Dataset | Bias (m) | RMSE (m) | MAE (m) | Lower decile (m) | Upper decile (m) |
|---|---|---|---|---|---|
| MERRA | -126.3 | 290.1 | 199.7 | -466.3 | 141.7 |
| MERRA2 | -48.3 | 172.3 | 88.4 | -194.6 | 16.9 |
| ERA–Interim | -67.5 | 215.0 | 119.5 | -281.3 | 33.3 |
| ERA–20C | -103.1 | 274.6 | 173.0 | -380.9 | 51.2 |
| CFSR | -114.9 | 262.8 | 192.0 | -422.0 | 143.1 |
| 20CR | -244.3 | 447.1 | 337.6 | -733.8 | 151.8 |
| JRA–55 | -131.9 | 272.4 | 199.2 | -439.9 | 123.5 |
| AIRS | -132.3 | 274.2 | 200.3 | -440.1 | 120.2 |
| MAR | -13.4 | 94.1 | 37.2 | -56.0 | 18.0 |



**Table 3: Trends (°C per decade) of ice sheet areal average annual mean SAT for given periods. For the CMIP5 ESMs, the trend of the ensemble mean and the median of the individual trends are shown, along with the number of positive, negative and significant trends. Bold type indicates significance at the 0.05 level.**

| Dataset | Linear trends (°C decade⁻¹) | | | | | |
|---|---|---|---|---|---|---|
| | 1901–1930 | 1931–1960 | 1931–1990 | 1990–2005 | 1990–2014 | 1901–2000 |
| MERRA | | | | **1.495** | **0.823** | |
| MERRA2 | | | | **1.739** | **0.696** | |
| ERA–Interim | | | | **1.979** | **0.907** | |
| ERA–20C | 0.513 | 0.023 | **-0.183** | **1.637** | | 0.020 |
| CFSR | | | | **1.526** | **0.589** | |
| 20CR | **0.393** | -0.468 | **-0.295** | **2.042** | **1.045** | **-0.156** |
| JRA–55 | | | | **1.799** | **0.949** | |
| MAR–ERA | | | | **1.358** | **0.559** | |
| MAR–20CRv2c | 0.161 | -0.120 | **-0.156** | **1.649** | **0.806** | -0.061 |
| CRU | 0.411 | -0.118 | **-0.169** | **1.249** | 0.505 | 0.023 |
| Berkeley Earth | 0.628 | -0.334 | **-0.227** | **1.703** | **0.965** | 0.054 |
| GISTEMP | 0.361 | -0.197 | **-0.139** | **1.447** | **0.865** | 0.100 |
| NANSENSAT | -0.065 | -1.003 | **-0.497** | **1.066** | | **-0.239** |
| CMIP5: ensemble mean | 0.119 | 0.016 | **0.088** | **0.485** | | **0.098** |
| CMIP5: median | 0.081 | 0.007 | 0.086 | 0.407 | | 0.094 |
| CMIP5: number positive (significant) | 23 (3) | 16 (0) | 23 (7) | 27 (4) | | 30 (20) |
| CMIP5: number negative (significant) | 7 (0) | 14 (1) | 7 (0) | 4 (0) | | 1 (1) |





| Network | Abbreviation | Period | Reference |
|---|---|---|---|
| Danish Meteorological Institute | DMI | 1784-2013 | Cappelen (2014) |
| Greenland Climate Network | GC-Net | 1995-2014 | Steffen and Box (2001) |
| Ohmura 1987 | Ohmura87 | 1930-1965 | Ohmura (1987) |
| PROMICE | PROMICE | 2007-2015 | van As et al. (2012) |
| University of Utrecht Kangerlussuaq transect | K-transect | 2003-2015 | van de Wal et al. (2005) |

**Figure 1:** Map of study area and weather stations used in this work. Symbol types represent the different monitoring networks summarized in the inserted table.



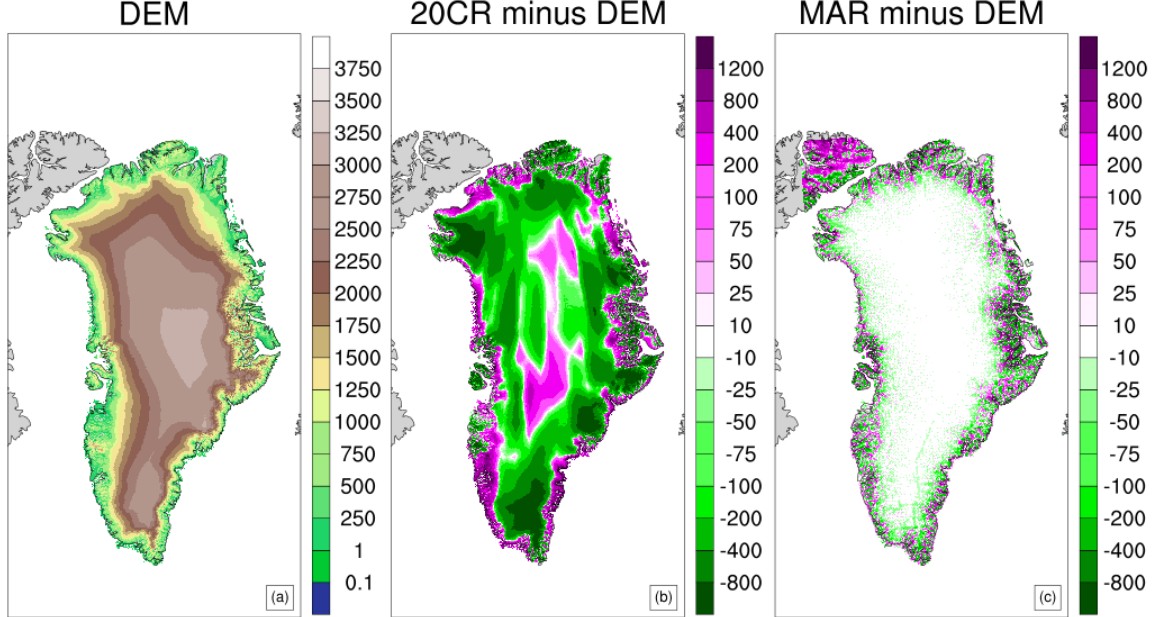

**Figure 2: (a) Digital elevation model (DEM) of Bamber et al. (2013) interpolated to EASE 5 km grid; (b) bias of 20CR surface elevation field interpolated to EASE grid, relative to Bamber et al. (2013); (c) bias of MAR surface elevation field relative to Bamber et al. (2013). Units are meters.**



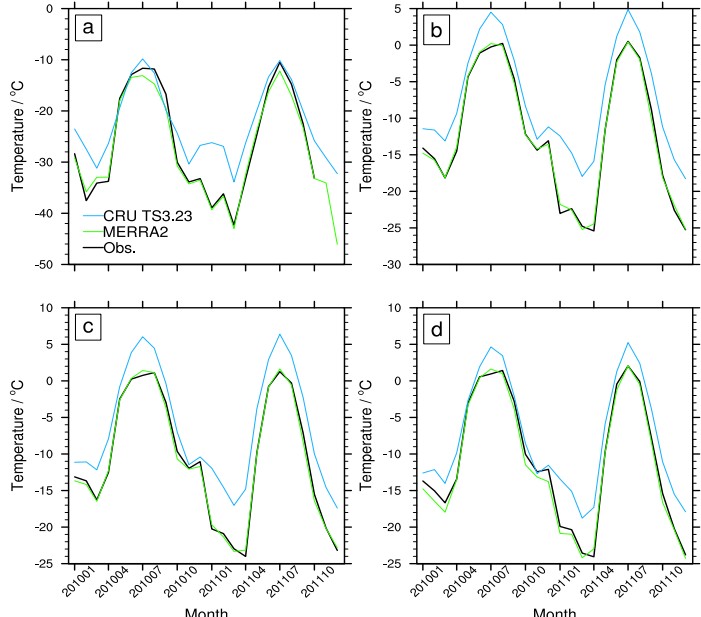

**Figure 3: SAT time series from four stations on the ice sheet for 2010 and 2011: (a) Summit (GC-Net) is at ~3200 m, close to the topographic summit of the ice sheet; (b) station S9 (K-transect) is at 1500m on the western flank of the ice sheet; (c) station KAN_M (PROMICE) is at 1270 m, close to the K-transect stations; (d) Swiss Camp (GC-Net) is at ~1200 m, north of the K-transect.**





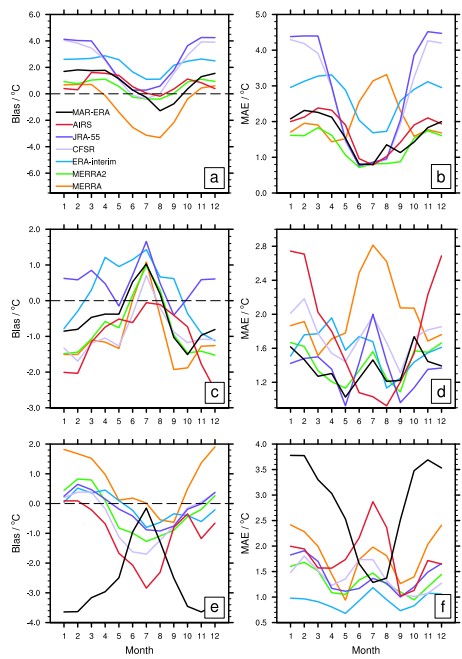

**Figure 4: Mean over station-months of bias (a, c, e) and absolute error (b, d, f) relative to monthly mean SAT at: ice sheet stations above 1500 m (a and b); ice sheet stations below 1500 m (c and d); and coastal (DMI) stations (e and f). Ice sheet stations are from GC-Net, PROMICE and K-transect. All available station months from 1979 onwards are used. All datasets included in this figure are elevation corrected: several shorter reanalyses, AIRS, and MAR-ERA. Note that the vertical scales vary with panels.**





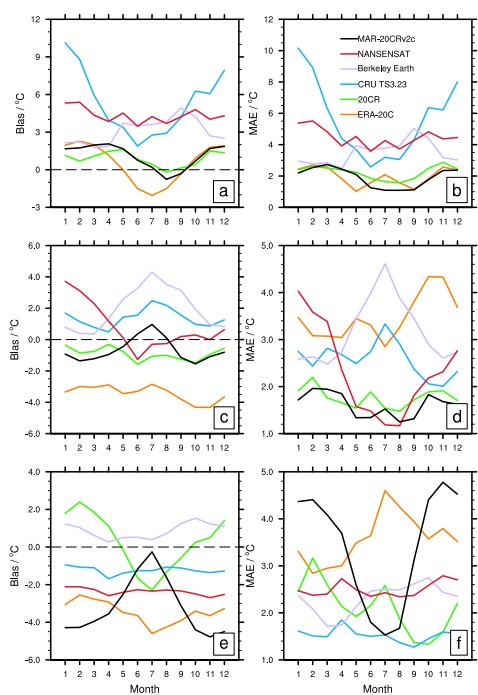

**Figure 5: As in Fig. 4, but for elevation-corrected long reanalyses and MAR-20CR, and three gridded SAT analyses (not elevation corrected).**





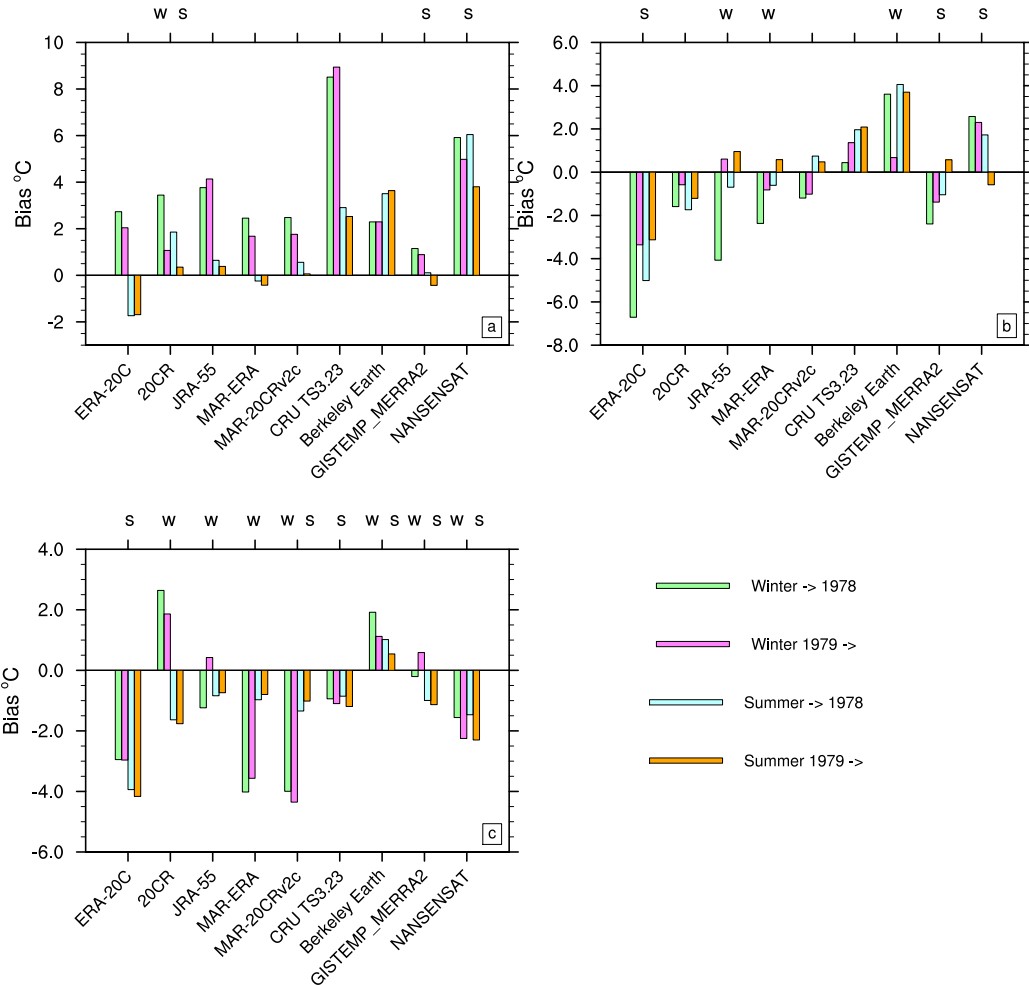

**Figure 6: Monthly mean SAT bias for winter (DJF) and summer months (JJA) before and after 1979, for all datasets that extend back before 1979 (elevation-corrected where applicable) at: ice sheet stations above 1500 m (a); ice sheet stations below 1500 m (b); and coastal (DMI) stations (c). Note that these are monthly SAT biases averaged over all months in a season, not biases of seasonal mean SAT. Changes significant at the 99% level (using Student's *t*-test with unequal variances) are denoted by w (for winter) and s (for summer) on the top axis.**



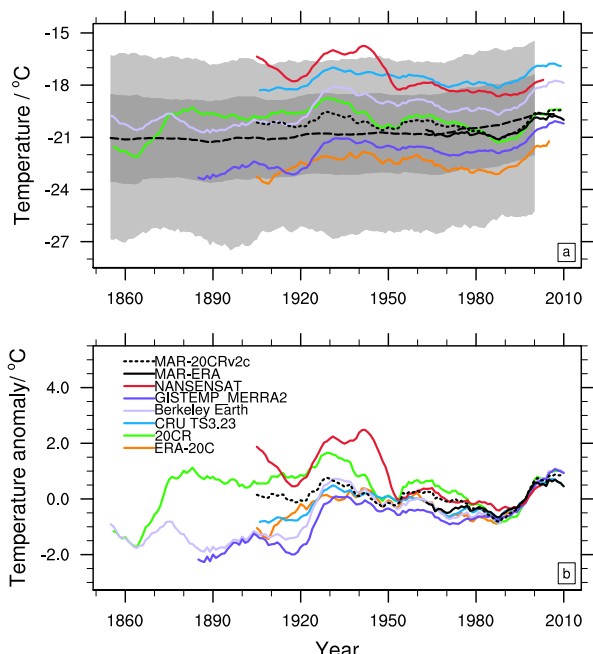

**Figure 7: (a) Time series of ice sheet areal average smoothed annual mean SAT for long reanalyses, gridded temperature analyses and both MAR variants (elevation corrected where applicable; colored lines) and CMIP5 climate models (not elevation corrected; ensemble mean in dashed black line; +/- 1 standard deviation in dark grey shading; maxima and minima in light grey shading). (b) Anomalies of ice sheet areal average smoothed annual mean SAT from long reanalyses, gridded temperature analyses and both MAR variants, relative to 1981-2010 mean. In both (a) and (b), time series are smoothed using a centered, uniform-weighted 11-year window, to highlight decadal variability and aid legibility.**