# Peer review of "Evaluation of Greenland near surface air temperature datasets"

_The Cryosphere, 2016_

## Short Comment (SC1) · 19 Jan 2017

The study brings together a large set of climate modeled surface air temperature output and compares the data with in-situ field observations at monthly temporal resolution.

The writing is clear.

major comments a.) The fact that the AIRS data are clear-sky means biases the more cloudy the all-sky retrieval is, as represented by field data. How large is that bias? It's seasonal range is low bias in summer and how large is the bias in winter, deg. C units?

b.) "Two regional data sets are also included", the the study neglects to compare field data with the Box (2013)* data that are worthy to compare because the 1.) span decades earlier than compared data; 2.) are in similar class of compared data GIS-

TEMP, BEST. 3.) p. 9 line 25 "the benefits for SAT of RCM downscaling are not clear" comparing with Box should help since Box mimics RACMO2.

The Box data are available at:

http://research.bpcrc.osu.edu/~jbox/Box_2013_Reconstruction_data/Box_Greenland_Temperature_monthly_1840-2014_5km_cal_ver20141007.nc

* Box, J. E. 2013. Greenland ice sheet mass balance reconstruction. Part II: Surface mass balance (1840-2010), Journal of Climate,Vol. 26, No. 18. 6974-6989. doi:10.1175/JCLI-D-12-00518.1

Along this line, the paper should compare these data along side the those in Figure S2.

p. 8, line 1 could include the Box data next to these others.

c.) Figure 3; selected sites are 3/4 W Greenland. Comparison more meaningful to select 1 site from NE Greenland, 1 SE, 1 Central, one SW or NW. So, recommend to keep Swiss Camp, use a KPC_ site, use a TAS site, use Summit, may be also use a THU_ site.

recommendation: Figure 3, scatter plots are desirable with dots represented by the month name

d.) you compare AIRS SAT retrievals, but why not also MODIS MOD11 as in the Hall et al work? You could justify not comparing with in-situ data because Hall et al already did. Then again, if you're after a comprehensive comparison and 'beauty contest', would be worth knowing which were more accurate AIRS SAT or MOD11.

e.) for more impact, seems worth more analysis of July or June through August temperatures since the headline-grabbing melt issue is more strongly tied to this part of the year. In discussion and conclusions, what novel melt season findings you deduce could get some attention through the frame of Greenland melting as a societal risk

factor.

f) try to more clearly distinguish spurious trends from real trends in 20CR and ERA 20C vs long term coastal DMI observations

minor comments

more clearly introduce NANSENSAT; appears abruptly p 7 line 14 without introduction

p. 3, line 26 remove "heavily"; adverbs are vague and unnecessary 4, line 17 remove "around"; unnecessary p. 7, line 7... "we are interested in changes in bias"... please more clearly frame why that is in your methods section. p. 11, line 6 "improved performance"; quantify that statement with a number, i.e. an x % reduced MAE and/or a x increase in correlation and/ or x reduction in bais. consider to rank tabulated MAE values to more clearly display which datasets are most accurate

---

## Referee Comment (RC1) · X. Fettweis (Referee) · 23 Jan 2017

This paper presents an interesting evaluation of the ability of a lot of reanalyzes, RCM and observations based datasets to represent the current near-surface temperature variability over Greenland. The paper is well written and deserves to be published in TC.

In addition to the Jason Box's suggestions (that I fully support), I have also several recommendations before publication:

- pg1, line 8: The MARv3.5.2 model should be explicitly cited in the abstract.

- pg2, line 9-11: the sensitivity of the MAR results to the reanalysis used as forcing has already been discussed in depth in

Fettweis, X., Box, J. E., Agosta, C., Amory, C., Kittel, C., and Gallée, H.: Reconstructions of the 1900–2015 Greenland ice sheet surface mass balance using the regional climate MAR model, The Cryosphere Discuss., doi:10.5194/tc-2016-268, in review, 2016.

and in particular for the simulation of the near-surface temperature (see Fig 7 of Fettweis et al., 2016). This works should be cited here and one of the goal of the present work is rather to extent the analysis of Fettweis et al. (2016) because a part of the proposed aim has already been made in Fettweis et al. (2016).

- pg 2, line 19: it is the version 3.5.2 which is used here. Fettweis et al. (2016) must be cited here. Why do not include also MARv3.5.2 forced by ERA-20C in this evaluation ? Outputs are also available on my FTP. In Table 1, change MAR-20CR by MAR-20CRv2c.

- pg 6, lines 14-20. I agree with the authors that Summit is a good candidate to evaluate the near-surface temperature. But, afterward, evaluation is done over all stations. Therefore, I don't see the interest of this small paragraph and of Fig3 which could be put in Supplementary material.

- pg 9, lines 17-25. This paragraph must be reformulated. Firstly, it is true that MAR is worse than reanalyses at the coastal stations but these DMI observations have been assimilated into the reanalysis and not into MAR! Secondly, MAR has been developed for well representing the ice sheet conditions and not the oceanic conditions. Sea ice cover and SST are forced into MAR because their impact on the SMB is low (Noel et al., 2014, TC). When there is sea ice, the sea ice thickness is prescribed to 1m in MAR and MAR computes itself its surface temperature. This explains why the results of MAR at the coastal stations (fully impacted by the neighborhood ocean conditions) are particularly bad in winter because using a fixed sea ice thickness of 1m overestimates the real sea ice thickness in most of the places and therefore allows a extreem cooling of the surface temperature in MAR. In summer, there is less sea ice, coastal nearsurface temperatures is more impacted by the SST (which is prescribed into MAR) and therefore the MAR results are better (because it is more constrained). This issue with sea ice in winter has been corrected in part in MARv3.6.

- pg 12, lines: 13-14: Done in Fettweis et al. (2016).

- All the datasets should be evaluated also at the summer (JJA) time scale as already suggested by Jason Box. It should be good to have the equivalent of Fig 7 for JJA.

- The mean biases for all CMIP5 model should be listed in Supplementary. What are the best CMIP5 models ?

- In addition to Table 3, mean bias as well as RMSE over 1980-2010 of all data sets used here should be listed for both ice sheet and coastal weather stations.

---

## Referee Comment (RC2) · Anonymous Referee #2 · 4 Feb 2017

This is a timely and fairly novel comparison of Greenland near-surface air temperature (SAT) bias trends between various reanalysis/regional climate model and instrumental-based gridded SAT products and in situ meteorological SAT data, and an analysis of Greenland SAT trends over time since 1900.

The paper is clear and well-written, and makes some interesting and useful conclusions about the likely accuracy and reliability of the various gridded SAT products that are analysed. It provides a valuable contribution to the relatively limited literature base on this topic, especially given the recent plethora of available reanalysis and regional climate model products for Greenland (including several products spanning the first half of the twentieth century).

I would like to have seen a bit more spatial analysis of Greenland SAT trends, e.g.

[Figure]

gridded maps of trends from several different datasets, showing how the trends spatially vary, where they are most significant and reliable for different time periods, and summarising the differences and explanations in these.

page 2, line 3: I don't follow why "the benefit for SAT [of high resolution] is less clear [than for SMB]" Why should this be the case, given that melt and runoff strongly correlate with SAT which is much more directly elevation-dependent than accumulation?

p.2, l.17: "The LATTER SAT reconstruction was compared..."

p.2, l.29 remove comma after "SAT observations".

p.5, l.9 How were the 31 model configurations selected from the CMIP5 ensemble?

p.5, l.20: add that 20CR also has a positive bias in the central-most regions.

p.6, l.5 "MERRA2 closely matches the observed seasonal cycle at all stations" - does MERRA2 assimilate data from these meteorological stations and is this therefore a non-independent comparison and unsurprising result?

p.9, l.27: "variability in the downscaled RCM".

Figure 3: need to distinguish the two faint lines more clearly.

---

## Author Comment (AC1) · 28 Mar 2017

The response is given in the supplement.

Please also note the supplement to this comment:
http://www.the-cryosphere-discuss.net/tc-2016-297/tc-2016-297-AC1-supplement.zip

—————————————

---

## Author Comment (AC2) · 28 Mar 2017

The response is given in the supplement.

Please also note the supplement to this comment:
http://www.the-cryosphere-discuss.net/tc-2016-297/tc-2016-297-AC2-supplement.zip

—————————————————

---

## Author Comment (AC3) · 28 Mar 2017

The response is given in the supplement.

Please also note the supplement to this comment:
http://www.the-cryosphere-discuss.net/tc-2016-297/tc-2016-297-AC3-supplement.zip

---

## Author Response (AR1)

JEB Box

The study brings together a large set of climate modeled surface air temperature output and compares the data with in-situ field observations at monthly temporal resolution.

The writing is clear.

major comments

a.) The fact that the AIRS data are clear-sky means biases the more cloudy the all-sky retrieval is, as represented by field data. How large is that bias? It's seasonal range is low bias in summer and how large is the bias in winter, deg. C units?

The goal of including the AIRS data in this work are to assess its overall suitability for air temperature monitoring over Greenland. We feel that attributing the bias to different sources (e.g., clear-sky versus all-sky; local time of retrievals; biases from retrieval algorithms) is beyond the scope of this study, and that the assessments presented and comparison with previous results (Koenig and Hall, 2010) are valuable as is. Nonetheless we extend the discussion of AIRS in Sect. 3.3.

b.) "Two regional data sets are also included", the the study neglects to compare field data with the Box (2013)* data that are worthy to compare because the 1.) span decades earlier than compared data; 2.) are in similar class of compared data GISTEMP, BEST. 3.) p. 9 line 25 "the benefits for SAT of RCM downscaling are not clear" comparing with Box should help since Box mimics RACMO2. The Box data are available at: http://research.bpcrc.osu.edu/~jbox/Box_2013_Reconstruction_data/Box_Greenland_Temperature_monthly_1840-2014_5km_cal_ver20141007.nc

* Box, J. E. 2013. Greenland ice sheet mass balance reconstruction. Part II: Surface mass balance (1840-2010), Journal of Climate,Vol. 26, No. 18. 6974-6989. doi:10.1175/JCLI-D-12-00518.1 Along this line, the paper should compare these data along side the those in Figure S2. p. 8, line 1 could include the Box data next to these others.

We thank the referee for making this data available, and include results from the dataset in Figs. 4, 5 and 6, and Tables 1, 3 and 4. Discussion of the results is also included.

c.) Figure 3; selected sites are 3/4 W Greenland. Comparison more meaningful to select 1 site from NE Greenland, 1 SE, 1 Central, one SW or NW. So, recommend to keep Swiss Camp, use a KPC_ site, use a TAS site, use Summit, may be also use a THU_ site. recommendation: Figure 3, scatter plots are desirable with dots represented by the month name

We thank the referee for this recommendation and alter the stations included in Fig. 3 accordingly. However, we retain the figure format as time series, in order to give an impression of the seasonal cycle and the effect that dataset biases have on their representation of seasonal cycle. This figure has now been moved to the supplementary material.

d.) you compare AIRS SAT retrievals, but why not also MODIS MOD11 as in the Hall et al work? You could justify not comparing with in-situ data because Hall et al already did. Then again, if you're after a comprehensive comparison and 'beauty contest', would be worth knowing which were more accurate AIRS SAT or MOD11.

We use AIRS and not MODIS for the reason that, nominally, AIRS includes a near-surface air temperature (SAT) product, while MODIS MOD11 is a surface temperature (LST) product. While this difference might seem somewhat artificial, we feel the introduction of further biases by comparing SAT with LST will reduce the value in including MODIS MOD11.

e.) for more impact, seems worth more analysis of July or June through August temperatures since the headline-grabbing melt issue is more strongly tied to this part of the year. In discussion and conclusions, what novel melt season findings you deduce could get some attention through the frame of Greenland melting as a societal risk factor.

Time series of summer mean SAT have been added to Fig. 6.

f) try to more clearly distinguish spurious trends from real trends in 20CR and ERA 20C vs long term coastal DMI observations

Time series of annual mean SAT from long term coastal DMI observations and (nearest grid point of) SAT datasets have been added to Supplemental Material.

minor comments

10   more clearly introduce NANSENSAT; appears abruptly p 7 line 14 without introduction
Brief introduction in Sect. 2.2.

p. 3, line 26 remove "heavily"; adverbs are vague and unnecessary
Done.

4, line 17 remove "around"; unnecessary
Done.

p. 7, line 7... "we are interested in changes in bias"... please more clearly frame why that is in your methods section.

20   We have clarified in Sects. 2.2 and 3.1 that stationarity of bias over time is relevant to the credibility of long term trends and variability.

p. 11, line 6 "improved performance"; quantify that statement with a number, i.e. an x % reduced MAE and/or a x increase in correlation and/ or x reduction in bais. consider to rank tabulated MAE values to more clearly display which datasets are

25   most accurate
Improvements due to elevation corrections have been quantified in Sect 4.
This paper presents an interesting evaluation of the ability of a lot of reanalyzes, RCM and observations based datasets to represent the current near-surface temperature variability over Greenland. The paper is well written and deserves to be
10  published in TC.

In addition to the Jason Box's suggestions (that I fully support), I have also several recommendations before publication:

- pg1, line 8: The MARv3.5.2 model should be explicitly cited in the abstract.

Due to the large number of datasets used, specific examples are only mentioned in the abstract in relation to key conclusions. Citing MAR would require citing all other datasets too, which we feel would be inappropriate for the abstract.

- pg2, line 9-11: the sensitivity of the MAR results to the reanalysis used as forcing has already been discussed in depth in
20  Fettweis, X., Box, J. E., Agosta, C., Amory, C., Kittel, C., and Gallée, H.: Reconstructions of the 1900–2015 Greenland ice sheet surface mass balance using the regional climate MAR model, The Cryosphere Discuss., doi:10.5194/tc-2016-268, in review, 2016.
and in particular for the simulation of the near-surface temperature (see Fig 7 of Fettweis et al., 2016). This works should be cited here and one of the goal of the present work is rather to extent the analysis of Fettweis et al. (2016) because a part of
25  the proposed aim has already been made in Fettweis et al. (2016).

We thank the referee for their important work on this subject and for making model outputs publicly available. Fettweis et al. (2016) has been cited and the results and conclusions have been factored into our work.

30  - pg 2, line 19: it is the version 3.5.2 which is used here. Fettweis et al. (2016) must be cited here.

Done.

Why do not include also MARv3.5.2 forced by ERA-20C in this evaluation ? Outputs are also available on my FTP.

This model run has now been included in Figs. 4, 5 and 6, and Tables 1, 3 and 4. Discussion of these results has also been added.

5    In Table 1, change MAR-20CR by MAR-20CRv2c.

Done.

- pg 6, lines 14-20. I agree with the authors that Summit is a good candidate to evaluate the near-surface temperature. But,
10    afterward, evaluation is done over all stations. Therefore, I don't see the interest of this small paragraph and of Fig3 which could be put in Supplementary material.

This section has been re-written and the relevant Figure has been altered (in line with J. Box's comments) and moved to the Supplementary Material.

- pg 9, lines 17-25. This paragraph must be reformulated. Firstly, it is true that MAR is worse than reanalyses at the coastal stations but these DMI observations have been assimilated into the reanalysis and not into MAR! Secondly, MAR has been developed for well representing the ice sheet conditions and not the oceanic conditions. Sea ice cover and SST are forced into MAR because their impact on the SMB is low (Noel et al., 2014, TC). When there is sea ice, the sea ice thickness is
20    prescribed to 1m in MAR and MAR computes itself its surface temperature. This explains why the results of MAR at the coastal stations (fully impacted by the neighborhood ocean conditions) are particularly bad in winter because using a fixed sea ice thickness of 1m overestimates the real sea ice thickness in most of the places and therefore allows a extreem cooling of the surface temperature in MAR. In summer, there is less sea ice, coastal near surface temperatures is more impacted by the SST (which is prescribed into MAR) and therefore the MAR results are better (because it is more constrained). This issue
25    with sea ice in winter has been corrected in part in MARv3.6.

This paragraph has been re-written to relay the explanation of coastal biases. However, the finding (that MAR performs worse than reanalyses at coastal stations) still stands and, we feel, should be reported. Further, to the best of our knowledge, it is generally not true that DMI (or any other network's) SAT observations are assimilated in reanalyses, with the exception
30    of ERA-Interim (in which they are used to update surface properties as part of its land surface analysis scheme; this may explain ERA-Interim's relatively low MAE at DMI stations). In addition, SAT observations *are* used in the gridded SAT analyses.

- pg 12, lines: 13-14: Done in Fettweis et al. (2016).

This sentence has been removed.

- All the datasets should be evaluated also at the summer (JJA) time scale as already suggested by Jason Box. It should be
good to have the equivalent of Fig 7 for JJA.

A summer (JJA) panel has been added to Fig. 6 (old Fig. 7).

- The mean biases for all CMIP5 model should be listed in Supplementary. What are the best CMIP5 models ?

The ice sheet annual mean SAT from CMIP5 EMSs are already listed in Table S2 and, alongside the same quantity from an
observational-based dataset, we think this gives an adequate idea of which CMIP5 ESMs are most and least biased. We
refrain from naming best CMIP5 ESMs as this is dependent on which comparison dataset is used – indeed, this fact is one of
our conclusions.

- In addition to Table 3, mean bias as well as RMSE over 1980-2010 of all data sets used here should be listed for both ice
sheet and coastal weather stations.

A new table has been added to showing mean bias and mean absolute error.
This is a timely and fairly novel comparison of Greenland near-surface air temperature (SAT) bias trends between various reanalysis/regional climate model and instrumental based gridded SAT products and in situ meteorological SAT data, and an

10    analysis of Greenland SAT trends over time since 1900.

The paper is clear and well-written, and makes some interesting and useful conclusions about the likely accuracy and reliability of the various gridded SAT products that are analysed. It provides a valuable contribution to the relatively limited literature base on this topic, especially given the recent plethora of available reanalysis and regional climate model products

15    for Greenland (including several products spanning the first half of the twentieth century).

I would like to have seen a bit more spatial analysis of Greenland SAT trends, e.g. gridded maps of trends from several different datasets, showing how the trends spatially vary, where they are most significant and reliable for different time periods, and summarising the differences and explanations in these.

We thank the referee for this recommendation and include maps of trends in the supplementary material.

page 2, line 3: I don't follow why "the benefit for SAT [of high resolution] is less clear [than for SMB]" Why should this be the case, given that melt and runoff strongly correlate with SAT which is much more directly elevation-dependent than

25    accumulation?

Our point here is that, because SAT is strongly elevation-dependent (as are melt and runoff, as you point out), the benefit of a high resolution model over a low resolution model with post-processing (e.g., the elevation corrections used in Hanna et al. (2011), Lucas-Picher et al. (2012) and our study) is not clear. On the other hand, for precipitation (and therefore SMB), high

30    resolution modelling appears to lead to improvement over low resolution modeling. This has been clarified in the text.

p.2, l.17: "The LATTER SAT reconstruction was compared..."
This sentence has been clarified.

p.2, l.29 remove comma after "SAT observations".

Done.

p.5, l.9 How were the 31 model configurations selected from the CMIP5 ensemble?

The model configurations were chosen based on availability of necessary data: SAT from historical runs and ice fraction fields. This has been clarified in the manuscript.

p.5, l.20: add that 20CR also has a positive bias in the central-most regions.

Done.

p.6, l.5 "MERRA2 closely matches the observed seasonal cycle at all stations" - does MERRA2 assimilate data from these meteorological stations and is this therefore a non-independent comparison and unsurprising result?

To the best of our knowledge, the MERRA2 reanalysis does not assimilate SAT from any land stations (in fact, of the reanalyses used here, only ERA-Interim assimilates SAT, which it uses to update surface properties as part of its land surface analysis scheme; this may explain ERA-Interim's relatively low MAE at DMI stations).

p.9, l.27: "variability in the downscaled RCM".

Done.

Figure 3: need to distinguish the two faint lines more clearly.

Done. (Note this figure has been moved to the supplementary material.)

[revised manuscript text omitted]

---

## Author Response (AR2)

**Responses to reviewers' suggestions (05 May 2017)**

Further to the changes presented in earlier revisions (tc-2016-297-author_response-version1.pdf), minor changes have been made to address the reviewers' suggestions.

5

Reviewers' comments are reproduced in black.

Authors' responses are in blue.

Changes to the manuscript are shown in red.

10 **Report #1 (Anonymous referee #2, 25 April 2017)**

page 12, line 12: I understood that GC-Net (inland Greenland) AWS data had recently been incorporated into the Global Telecommunications System and are therefore assimilated into some (ECMWF?) reanalysis - please check and modify this sentence if necessary.

15 Some GC–Net stations are now indeed included in WMO data broadcasts on the Global Telecommunications System. However, it appears that they have only been included since approximately April 2016, and so do not affect our analysis (which includes GC–Net data up to the end of 2014). Even though the GC–Net stations are now available for assimilation into ERA–Interim, it remains unclear whether they in fact are, or if they are "blacklisted". We thank the referee for bringing this to our attention.

20

The text in red has been added to the relevant sentence: "To the best of our knowledge, for the period analysed here the only Greenland SAT observations that are assimilated by ERA–Interim are from DMI stations, and so the ice sheet stations still provide independent data."

25 **Report #2 (Xavier Fettweis, 03 May 2017)**

Minor: please update the reference Fettweis et al. (2016) to the TC version: Fettweis, X., Box, J. E., Agosta, C., Amory, C., Kittel, C., Lang, C., van As, D., Machguth, H., and Gallée, H.: Reconstructions of the 1900–2015 Greenland ice sheet surface mass balance using the regional climate MAR model, The Cryosphere, 11, 1015-1033, doi:10.5194/tc-11-1015-2017, 2017.

30

The citations and reference have been updated accordingly.